# Patient Perspectives on the Care in a Long COVID Outpatient Clinic—A Regional Qualitative Analysis from Germany

**DOI:** 10.3390/healthcare13070818

**Published:** 2025-04-03

**Authors:** Lea Alexandra Gölz, Regina Poß-Doering, Uta Merle, Michel Wensing, Sandra Stengel

**Affiliations:** 1Department of Primary Care and Health Services Research, Heidelberg University Hospital, Heidelberg University, 69120 Heidelberg, Germany; leagoelz@posteo.de (L.A.G.); regina.poss-doering@med.uni-heidelberg.de (R.P.-D.); michel.wensing@med.uni-heidelberg.de (M.W.); 2Department of Internal Medicine IV, Heidelberg University Hospital, Heidelberg University, 69120 Heidelberg, Germany; uta.merle@med.uni-heidelberg.de

**Keywords:** post-COVID, outpatient clinic, specialist care, health systems research, primary care

## Abstract

**Background/Objectives**: Long COVID specialized outpatient clinics (sOCs), which are part of the recommended long COVID care, usually face high demand. Few studies focused on the experience of care in such facilities in Germany. This study investigated how patients experience care in a sOC at a German university hospital. **Methods**: Semi-structured interviews were conducted with patients attending this clinic between October 2022 and January 2023. Data analysis was based on thematic analysis. **Results**: The themes from interviews with 14 patients (F = 11, M = 3) could be broadly categorized into statements on the pathway to the sOC, and statements on care provided in the sOC. Findings show that patients’ high expectations at the sOC appointment were shaped by previous experiences with care, which were mainly perceived as inadequate. Care in the sOC was predominantly perceived as competent, empathetic and relevant for further care and coping with the disease. A deterioration in health directly related to the consultation (classifiable as post-exertional malaise) was frequently described, as was a high need for ongoing consultation. **Conclusions**: Overall, the findings point to a need for adaptations in the sOC, such as identifying optimized models of care and tailoring them to the patients’ limited resources. This includes measures to improve care outside the sOC.

## 1. Introduction

Long COVID (LC) is characterized by symptoms that continue or develop after the initial SARS-CoV-2 infection and last more than four weeks [1,2]. The term ‘long COVID’ is not used consistently in studies or frameworks [1]. For example, according to the World Health Organization’s more specific clinical case definition [3], the post-COVID condition describes symptoms that usually occur three months after symptoms of SARS-CoV-2 infection and cannot be explained by any other diagnosis. Symptoms of LC can affect the lungs, heart, immune system, nervous system and other organ systems, with fatigue, cognitive impairment and psychological disturbances such as anxiety, depressive levels and sleep disturbances being the most commonly reported [4]. There is also evidence that myalgic encephalomyelitis/chronic fatigue syndrome (ME/CFS) may develop as a particularly severe form of LC, which, in addition to fatigue, typically has a worsening of symptoms after exercise, known as exercise intolerance, as a criterion [5,6,7,8,9,10,11,12,13,14]. Knowledge about LC is currently accumulating rapidly [15]. Nevertheless, much remains to be learned about its pathogenesis, treatment, and how best to optimize care [16], providing ongoing challenges for patients, health care givers, and healthcare systems worldwide [17,18,19,20,21,22,23].

Most patients with LC experience impairments in their daily function, quality of life and participation in social and work activities [14,24,25,26,27,28]. It can therefore be assumed that LC has a significant effect on the health of the population [26]. In addition, those affected face stigma [29]. It also has consequences for the economy: the gross value-added loss due to LC is estimated at 5.7 billion euros for Germany [16,30].

A population-based study conducted in Germany reported that 6.5% of COVID-19 patients may have new symptoms six to twelve months after SARS-CoV-2 infection, leading to at least moderate impairment of daily life and reduced health or working capacity [31]. The prevalence found is consistent with international findings [16].

Owing to its still unclear pathogenesis, the diagnosis of LC is based on certain criteria related to symptoms and the exclusion of other causes [1,32]. Therapeutic evidence also remains limited [33], leading to a lack of specific therapies, especially symptom-based approaches [5,34,35]. There is uncertainty about recommendations for exercise and activity in LC and ME/CFS. Study results show heterogeneous results with a not very robust study situation [36,37]. On the one hand, study results suggest improvement with rehabilitation, but on the other hand, there is also evidence of worsening of symptoms in terms of post-exertional malaise in subgroups with overexertion. [8,34,36,37,38,39,40,41,42].

Nationally and internationally, multidisciplinary and cross-sectoral care is recommended to address this complex illness entity [14,18,43,44,45,46]. In Germany, the guidelines for long COVID [14,47] call for a stepped care approach: general practitioners (GPs) are the first point of contact for patients and provide basic care in the sense of a generalist approach, whereas specialized outpatient clinics should be available for severe and complex cases [46]. The latter continue to be in high demand in Germany [48] as well as internationally [1]. The tiered approach can also be found in other countries [1].

The challenges of managing LC in primary care, patients’ perceptions of inadequate care, and the need for the integration of specialized care for LC have been extensively studied internationally [17,18,22,23,29,39,49,50,51,52,53,54,55]. To the best of our knowledge, research on the patient experience of LC care in specialized care settings is limited [48]. Therefore, the aim of this study was to explore patients’ perspectives on their care in a specialized long COVID outpatient clinic (sOC) at a university hospital in Germany.

## 2. Materials and Methods

### 2.1. Study Design

A cross-sectional, exploratory qualitative study design [56,57] with problem-centered, semi-structured interviews [58] was chosen to collect the subjective and individual experiences of patients in as much detail and differentiation as possible.

The reporting of the study is based on the Consolidated Criteria for Reporting Qualitative Research [59]; see Appendix A. Patients and the public were not involved in designing, conducting, or analyzing the study.

### 2.2. Recruiting and Study Participation

Between 10 October 2022, and 3 January 2023, study participants were recruited at a specialized long COVID outpatient clinic (sOC) of a university hospital in Germany, which is part of a regional long COVID network. The sOC services consisted of a single appointment that included the completion of questionnaires, tests (cognitive test, stress test, handgrip test), laboratory tests, and a doctor–patient consultation. Two specialists (internal medicine; general practice), a senior consultant, and a nurse were involved in the sOC. The patient and the general practitioner received a physician’s letter with further recommendations. If necessary, further referrals were made to other disciplines within the institution. Initially, purposive sampling was intended to account for variance in sex, age, place of residence and course of acute COVID-19 disease [60]. This strategy was changed to convenience sampling on 12 October 2022 following a review of scheduled appointments in the sOC in the following months and due to, therefore, anticipated difficulties in recruiting a sufficient number of participants in a reasonable time [60]. During the recruitment period, all patients coming in for their appointment were verbally informed about the study by the sOC staff or the first author and received an invitation letter and an information leaflet about the study. Patients who were interested in participation received more detailed information in person or by telephone from the first author.

The participants gave their written informed consent and were reimbursed EUR 50 (funded by the Ministry of Science, Research and Arts of the State of Baden-Wuerttemberg, no funding code).

Recruitment was scheduled to stop when saturation could be assumed. Based on field notes, a comparison of interview content aspects and literature on saturation and sample sizes in qualitative research was conducted [61,62]. Thematic saturation was to be assumed when interviews had provided extensive information about the questions and aims of the study and only minimal new information could be collected through further interviews. Thus, and given the homogeneity of the study group, thematic saturation was assessed and recruitment was stopped on 3 January 2023.

### 2.3. Data Collection

An anonymized written sociodemographic questionnaire was used to collect information about the personal and life situations of the participants (age, gender, place of residence, occupation, partner, care work), their COVID-19 experience (time and course of the disease), their inability to work due to LC, their relationship with their GP, and their functional status according to Klok et al. [63] before COVID-19 and at the current time.

Based on the theoretical background and current research, a semi-structured interview guide was developed through a process of collecting, reviewing, sorting and subsuming appropriate questions [64,65]. The thematic expertise of the first author, the methodological expertise of the second author, and the qualitative and clinical expertise of the last author were incorporated into the formulation of the questions. Owing to the high stress levels of the target group, a separate piloting of the interview guide was deemed challenging and potentially unethical. Therefore, the guide was piloted for comprehensibility and practicality in the first two interviews. Subsequently, in collaboration with the first and second author, the wording of specific passages where comprehension issues arose in the interviews was slightly adjusted.

The following topics were covered by the guide: the diagnosis of LC, access to sOC, experiences with and in the sOC, knowledge and experiences of the regional long COVID network, and the recommended stepped care approach (Appendix A).

The participants individually chose the mode of interview based on their personal preference and capacities: telephone interview, video conference via the web conferencing service of Heidelberg University’s computer center, face-to-face interview at the university hospital or in their personal home environment. Participants had the option of taking a break from the interview or continuing the interview on another day in order to be consistent with their individual capacity and limits and to avoid being overloaded by their participation in the study. No one other than the interviewer or the participant was present during the face-to-face interviews, and no one else was present during the telephone and video conference interviews, if this could be determined. The first author conducted all interviews and took field notes in all cases. All interviews were recorded with a digital, portable voice recorder, pseudonymized and transcribed verbatim using f4 transkript Edu version 7.0.6 software in accordance with the institutional transcription guidelines. The transcripts were not submitted to the participants for correction or comment.

### 2.4. Data Analysis

The sociodemographic data were analyzed via descriptive statistics via Microsoft Office Professional Plus 2019 Excel software.

The qualitative data were organized and evaluated by the author according to Braun and Clark’s reflexive thematic analysis [65], first on paper and then via MAXQDA Plus 2018 software (Release 18.2.4, Build 200406). Reflexive thematic analysis [65] was chosen because of its flexibility allowing both inductive coding, in order to capture the patterned meaning of the largely unknown individual patient experiences with their care in the sOC, and deductive coding, to reflect the problem-centered nature of the interviews. The six phases of thematic analysis (familiarizing with the data set; coding; generating initial themes; developing and reviewing themes; refining, defining, and naming themes; writing up) served as a guide for data analysis, which was not seen as a linear process, but as an iterative one that could be adapted as needed.

Familiarity with the data was achieved through intensive and reflective engagement with the data and the creation of notes and case summaries [65]. The first phase of data analysis took place concurrently with recruitment and data collection. The participants’ individual experiences and perspectives of LC-related care were categorized into patterns.

The coding of three interviews was initially paper-based and inductive to identify initial relevant aspects from the material. To take into account the problem-centered nature of the interviews and the corresponding orientation of the interview guide, deductive coding was then carried out until no further new codes were needed to capture relevant aspects [66]. Initially, coding was mainly semantic [65] and then supplemented by latent coding [65] to capture implicit statements. On the basis of the identified aspects, initial themes related to the research question were identified, applied to the entire dataset via MAXQDA Plus 2018 and differentiated several times in an iterative process (Appendix A).

### 2.5. Reflexivity

The interviews were conducted and analyzed by the first author as part of her master’s thesis (Bachelor of Science in Occupational Therapy and little experience in qualitative research). All steps of the analysis were methodically supervised, and the themes and codes generated were discussed by a senior researcher from the study team (SS, experienced in qualitative research, and a physician in the sOC) in regular team meetings. In research workshops held alongside the mast, the analysis steps and interim results were also discussed with junior and senior researchers from the study team (RPD-health services researcher, experienced in qualitative research; MW-health services researcher with expertise in qualitative research). The participants were informed that the interviews and primary data analysis were being conducted as part of a master’s thesis by a master’s student who was not involved in providing care in the sOC and that there was no relationship between the participants and the first author prior to the study.

## 3. Results

### 3.1. Sample Description

Fourteen of the twenty-nine invited people with LC, mostly females and middle-aged individuals, took part in the study (Table 1 and Table 2). Acute COVID-19 was an average of 10.4 months prior to the study, and participants were predominantly moderately functionally impaired compared to the pre-COVID-19 situation where they were predominantly functionally unimpaired (Table 2). One participant reported being additionally ill by the effects of immunization against SARS-CoV-2. To maintain the pseudonymization of this participant, given the subject matter of this paper, no statements related to this theme are included below. Ten interviews were conducted by telephone, three by videoconference and one in person at the Department of Primary Care and Health Services Research of the University Hospital Heidelberg. The interviews lasted on average 53 min (min. 23 min, max. 98 min). At the participants’ request, two interviews were split into two sessions, one of which was continued the following day and one after 28 days. Short breaks (2–5 min) were included in two interviews at the participants’ request.

The results are reported descriptively along two key themes identified from the data: pathway to the specialized long COVID outpatient clinic, and care provided in the specialized long COVID outpatient clinic. Table 3 provides an overview of the identified key themes and sub-themes. The theme definition and anchor quotations can be found in the appendix (Appendix A). Participants’ quotes have been carefully translated and are cited by interview number and line number.

### 3.2. Pathway to the Specialized Long COVID Outpatient Clinic

#### 3.2.1. Care Outside the Specialized Long COVID Outpatient Clinic

Access to healthcare providers

The majority of participants stated that access to providers from different groups and sectors was characterized by waiting times of several months. Some participants expressed not only understanding but also helplessness and disappointment at delays in diagnosis and further care.

*‘[…] you cannot get appointments with specialists anyway, so it does not go any further’*.(I03, line 221ff)

Care experienced as inadequate

Some participants reported that they had experienced a lack of competence in LC among healthcare providers from different professional groups and sectors. The perceived lack of competence was often associated with their own resentment, uncertainty about the further care process, and a worsening of their condition, as the expected help did not materialize.

*‘I was actually very helpless because my GP did not know what to do, I did not know what to do, my physiotherapist did not know what to do >laughs<’*.(I09, line 61 f)

Some participants expressed dissatisfaction with diagnostic and therapeutic interventions, describing overload reactions. Some stated that they felt left alone with their complaints and not taken seriously by general practitioners, other specialists, or inpatient rehabilitation. The latter was also described as *’medical gaslighting’* (I11, line 313).

A few participants reported that they did not feel that their symptoms could be consistently attributed to mental or psychosomatic illness in different care contexts.*‘I was also a little scared [at the sOC appointment], because I had heard from many doctors before, especially in rehab, that they think it is more of a mental thing and actually all long COVID patients should go to psychosomatic medicine, because we cannot truly find anything’*.(I09, line 67 ff)

Openness to long COVID

Some participants reported a willingness on the part of healthcare providers to be involved in LC-related care. They stated that this was due to, among other things, carrying out an exclusionary diagnosis, issuing prescriptions and referrals, and feeling taken seriously and encouraged.

#### 3.2.2. Own Initiative and Personal Responsibility

The majority of participants reported that they were actively involved in their LC care, for example, by obtaining up-to-date information and advice on LC and passing it on to GPs, taking a leading role in decisions about exclusionary diagnosis and specialist referrals, and vocational rehabilitation. According to the participants, the idea of visiting the sOC came mainly from themselves. Overall, some steps in the care process were initiated by the participants, and the roles of patient and doctor were thus perceived as *‘reversed’* (I04, line 71 f).

The reasons given by the participants for their own initiative and the responsibility it entailed were the perceived lack of competence of the care providers, their dissatisfaction with previous care, and their desire for specialist advice in the face of symptoms that in some cases had been increasing for months.

The participants’ perceptions of their own responsibilities varied widely. On the positive side, they felt that by taking the initiative, they could rule out more dangerous illnesses and avoid deterioration due to potentially harmful treatment options.

*‘[…] the rehab […] would definitely have had the potential to make me worse if I hadn’t already read a lot about it myself’*.(I12, lines 53 ff)

It was criticized that personal initiative and responsibility required energy and resources when the participants were *‘already weakened by their illness’* (I04, line 66). Notably, there are numerous sources of information of varying quality: the line between scientifically sound and nonserious information is *‘very, very thin’* (I12, line 308), and it is difficult to understand and implement recommendations for action without professional support.

*‘[…] but that is not the goal, that […] the patient has to be so well informed that he knows exactly which specialists he needs to go to’*.(I11, lines 335 ff)

#### 3.2.3. Appraisal of the Stepped Care Approach

The majority of participants experienced a stepped care approach in their care process. In general, participants welcomed the primary care provided by GPs, who were seen as the *‘linchpin’* (I04, line 448) of care, who know you and are available.

*‘I mean, in terms of capacity, and, and on top of that, the GP is on site. (…). The GP knows you beforehand and can judge you quite well’*.(I03, lines 283 ff)

However, the acceptance of this role for GPs in the context of LC was made contingent on conditions that were described as inadequate in current care. GPs were criticized for their perceived lack of competence and capacity to care for LC patients.

*‘But then, I’m saying now, the general practitioners must also be obligated to deal with the subject’*.(I10 lines 244 ff)

Finding a suitable GP and thus receiving adequate primary care was therefore a matter of *‘luck’* (I09, line 235) or *‘bad luck’* (I14, line 310). The idea of an adaptive process in the care structure also came up:

*‘In principle, it belongs there, with the idea that in ten years we will know as much about LC as we do about asthma and the common cold, and that it will be part of the standard repertoire of GP medical education. But we are not there yet’*.(I14, lines 304 ff)

### 3.3. Care Provided in the Specialized Long COVID Outpatient Clinic

#### 3.3.1. Expectations of the Appointment at the Specialized Long COVID Outpatient Clinic

Participants’ expectations of their appointment at the outpatient clinic related to comprehensive interdisciplinary diagnostics, including diagnosis and the issuing of a medical report; the implementation of specific therapies, including participation in drug trials or other experimental therapies; and the management of further personal care.

The sOC was seen as an institution with distinct expertise and a superior position to general practitioners and other specialists and was of high personal relevance.

The sOC ‘*was ultimately the last hope you could cling to, because there was nothing else*’. (I14, line 95 f)

Some participants had no expectations of their appointment in the sOC because they did not feel able to cope with it due to their illness or because they were disappointed with the care they had received.

*‘[…] I did not have any expectations because I was simply disappointed with the care situation in Germany anyway’*.(I13, line 46 f)

While participants reported that their expectations were largely met in general, those regarding long-term follow-up and participation in drug or other therapy trials were not.

In addition, the expectations of other patients hoping for therapies to cure LC from sOC were described as unfulfillable. The participants noted that it is important to consider such expectations to avoid disappointment.

#### 3.3.2. Process of Appointment at the Specialized Long COVID Outpatient Clinic

Waiting time for the appointment at the specialized long COVID outpatient clinic

According to the participants, the time between the first attempt to contact the sOC and the appointment ranged from four and a half to eleven months. Few participants accepted the waiting time or would have expected it to be longer. Others perceived it negatively, for example, as *’too long’* (I08, line 192) or *’very disappointing’* (I05, line 162), and described a delay in the care process and a deterioration in their health during the waiting period.

*‘If I had come to you relatively early […] you could have reacted much earlier. And for me, it was like nine lost months’*.(I10, line 235)

Course and duration of the appointment at the specialized long COVID outpatient clinic

From the participants’ point of view, the process of appointment at the sOC was made more difficult by the need to visit several diagnostic departments, the distances involved and the waiting times and was therefore also described as *‘a pain in the ass’* (I07, line 229). Other participants described the process as orderly and structured, understood temporary *’confusion’* (I09, line 94) in the hospital procedures, or did not perceive waiting times as long.

With a total of four to six hours, the appointment usually lasted longer than the participants expected and was compared *to ‘marathons’* (I04, line 617). Breaks between different examinations and discussions and opportunities to lie down were seen as positive but also as insufficient. One positive aspect was that information about the condition, questionnaires, and tips on how to prepare for the doctor’s consultation were emailed to participants prior to the appointment, which was perceived as reassuring.

#### 3.3.3. Experienced Competences

The participants perceived a *‘bundling of competences’* (I14, line 147) in the sOC due to the LC-related knowledge, its application and the professional attitude of the staff. Reference was given to the diagnostics carried out, the options and recommendations for therapy discussed, and the communication and interaction with the participants.

*‘I found it very, very competent, dealing with long COVID, I had the impression that you are understood as a long COVID patient, that you simply have the chance to be heard, that these are people who are naturally familiar with the clinical picture, how you feel. There was also a lot of attention given to how you felt, and diagnostics were also carried out’*.(I06, line 272 ff)

The participants welcomed this knowledge and expertise and therefore felt *‘very relieved’* (I03, line 103), *‘in good hands’* (I04, line 342) and *therefore ‘naturally also felt safe and confident’* (I06, lines 190 f).

#### 3.3.4. Differences from LC Care Outside of the Specialized Long COVID Outpatient Clinic

While few of the participants considered the measures taken in the sOC to be implementable in the general practitioner setting or explicitly described no differences between the care inside and outside the sOC, others described the systematic diagnosis, the well-founded therapy recommendations, an empathetic approach and the professional knowledge of sOC employees as key differences in care. The sOC has time, money and specific equipment that serves LC care and is lacking outside of the sOC.

*‘Therefore, I did not have to explain this to the doctor, but the doctor can now explain something to me about my illness. That was a big difference’*.(I03, line 121 ff)

#### 3.3.5. Consequences of Appointment at the Specialized Long COVID Outpatient Clinic

Direct health consequences

The sOC appointment was described as stressful, in part because of the duration of the appointment, the large number of perceived stimuli and stress caused by the diagnostic measures, and the reduced ability to cope with stress due to illness.

*‘(…), because any kind of stress or tension causes weakness and exhaustion and confuses my nervous system’*.(I04, line 270 ff)

According to most participants, the effort led to an aggravation of existing symptoms and severe exhaustion and even overload so that, in some cases, not all diagnostic measures could be carried out or the participants were unable to actively participate in the discussion with the doctor because of the preceding procedures.

*‘[…] so the whole day was a complete border crossing for me. The biggest one I’ve ever had’*.(I11, line 107 f)

*‘That’s just because, I had afterwards, it really took me more than a week, I really had such a weakness afterwards, I was really bad. So, because it’s, obvious, so many hours of such high tension and exertion had to, had to lead to a complete crash’*.(I04, line 337 ff)

Post-exertional malaise may result in the need to lie down or even the inability of patients to leave the clinic on their own.

*‘When I felt unwell, when I had a crash, they called my home and told them to come and pick me up’*.(I13, line 138 ff)

However, the exertion has sometimes had a positive connotation, in the sense of being ‘*glad this is being investigated*’ (I03, line 84)*,* or also because of the opportunity to show one’s limitations.

*‘But in retrospect, I actually thought it was pretty good that I was there on a day when there were so many symptoms, because it’s this, this fluctuating thing that makes it so hard to grasp’*.(I12, line 189 ff)

Consequences for further care

In terms of the subsequent care process, the participants attached great importance to the sOC doctor’s letter, as a written diagnosis from the sOC makes it possible to apply for support services and to declare LC as an occupational disease and was used in communication with other caregivers.

Among other things, the sOC’s recommendations for further diagnosis, psychotherapeutic support, self-management, and sociolegal support, as well as prescriptions for remedies and assistive devices, gave the participants clarity about what to do next.

*‘So, I have a roadmap and it is good that I now have something at hand and that it is also structured’*.(I06, line 243 ff)

#### 3.3.6. Single Appointment at the Specialized Long COVID Outpatient Clinic

Evaluation of one-time appointments at the specialized long COVID outpatient clinic

The current arrangement of the one-time appointment in the sOC did not correspond to the idea of continuous support but rather gave the participants the feeling of being left alone. Some participants found it helpful that it was still possible to contact the sOC by email or telephone afterward.

*‘[…] then you are alone again. So that is a basic principle of the illness, that you’re basically left alone. […]’*.(I03, line 98 f)

Possible reappointment at the specialized long COVID outpatient clinic

Some participants thought that a second appointment three to twelve months after the sOC would be useful if there were new findings, treatment recommendations or *‘recommendations for off-label or experimental drugs or treatments’* (I11, line 259 f.). It was assumed that a new consultation would not be necessary if the doctor *‘implements the recommendations of the post-COVID outpatient clinic well’* (I05, line 429 f).

#### 3.3.7. Indications for an Appointment at a Specialized Long COVID Outpatient Clinic

Most participants would advise other LC patients to make an appointment at the sOC *‘as early as possible’ (I08, line 192; I10, line 210*) up to a maximum of three months after the onset of symptoms in the case of persistent complaints or a reduced capacity to act.

According to the participants, the reasons for this recommendation include the support experienced in the sOC and that *‘you get the understanding that everyone wants’* (I05, line 302).

The participants noted that it was only possible to a limited extent to see other patients in the sOC owing to the low capacity of the sOC and the function of the sOC as a *‘temporary, scientific supporting institution’* (I14, line 272 f).

## 4. Discussion

### 4.1. Discussion of the Results

The aim of this qualitative study was to explore how patients experience LC care in the sOC of a university hospital in Germany. The main findings emerged in the two main themes ‘pathway to sOC’ and ‘care provided in the sOC’. Barriers to accessing care outside the sOC were revealed, including a perceived lack of competence in the healthcare system, underserved care, psychologization, and, as reported by some participants, openness on the part of caregivers outside the sOC. Evaluations of the stepped care approach revealed both potential strengths and, more importantly, perceived weaknesses. A predominantly high degree of necessary self-initiative was described, sometimes resulting in ‘reversed’ roles of patients and physicians. Expectations related to the sOC were mainly based on experiences outside the sOC, perceptions of continuing care in the sOC, participation in studies, and the awareness of a lack of specific therapies. Positive aspects of the sOC appointment included competence, empathy, and a systematic process of diagnosis and treatment recommendations, which had a positive impact on further care in terms of having a ‘roadmap’ and coping with the disease. The negative aspects included long waiting times, long and overburdening appointments, and the single appointment strategy.

Participants’ experiences with care outside the sOC are consistent with extensive findings in the literature [54], reflecting a consistent lack of knowledge among healthcare providers [1,16,18,22,29,51,67].

These problems are well known from the treatment of other post-acute infectious syndromes, such as ME/CFS, and persist here [67,68,69]. In a systematic review, von der Lippe et al. [70] reported that people with rare diseases also experience a lack of knowledge among health professionals, more often in primary care and less often in specialized settings, which can result in incorrect treatment, delayed diagnosis, and refusal of social services. Feelings of not being taken seriously, being left alone, and even being stigmatized were also identified as key aspects of care that were perceived as inadequate and are also known from other studies [67,68,69].

The theme ‘own initiative and personal responsibility’ aligns with previous studies in the LC context [18,51], highlighting their relevance. The role reversal of GPs and patients described by the participants can be discussed with respect to medical-sociological considerations regarding the roles of patients and doctors. The doctor is usually attributed to medical expertise [71,72,73,74,75]. The patient is associated with the use of medical services and can act as an expert for his/her personal experience of illness [76,77,78]. This regular understanding of roles contrasts with the roles that some participants in this study reported. Again, there are parallels with rare disease patients, whom von Lippe et al. [70] call patient experts, who often research on the internet, have more information about their illness than professionals do, and develop *“a feeling of being the best ones to make decisions about their diagnosis and its treatment”* [70] (p. 776). When patients perform tasks that are typically the responsibility of a physician, it could have both positive and negative effects on navigation, diagnosis, and therapy, including potentially a negative impact on patient safety. One way to use patient knowledge constructively would be to integrate it into tailored patient-centered interventions [79].

Regarding the sOC, high expectations are related to the experienced undersupply in the previous care context. However, the expectations are realistic and illustrate the knowledge of the current lack of established specific therapies for LC [16,33], and the limited capacity of comparable institutions, in keeping with the results of the literature [1,48].

The findings show that strengthening the primary care-based stepped care approach, which is already internationally practiced [1,47,54], is necessary for the acceptance of care outside the sOC and thus for care coordination. The reported limitation of sOC resources in Germany reinforces this idea [48]. At the same time, it seems important to develop further concepts for sOC that address the reported patients’ needs but also address the lack of evidence for evaluating the effectiveness of LC care models [1,16,39]. Lessons learned from other chronic care models can be incorporated into evaluation designs [80,81,82]. The perceived expertise and empathic behavior of the care providers as well as the perceived relevance of the appointment for further care management and for coping with the disease explain the criticism of a GP-based stepped wedge approach, which is experienced by most participants as inadequate, which is in line with the literature [83,84]. The knowledge and competence gaps in the field of LC identified in the study, which have also been documented internationally and persistently [22,29,85], highlight the urgent need for appropriate continuing medical education for healthcare professionals, which is also being addressed internationally [1,16]. In doing so, the challenges of ‘evolving knowledge’ need to be addressed [86]. In addition to training healthcare professionals, educational measures for patients could also help improve the quality of care [87] and thus close identified gaps in care. Networks could constitute another approach to increase the competence and motivation of primary care physicians in LC care and promote cross-sectoral knowledge transfer and collaboration [18,88]. This could include, for example, telemedicine consultations or boards [39]. A supporting element is described by Stallmach et al. in terms of mobile primary healthcare for post-COVID patients in rural areas [89].

A central aspect that emerged from the interviews is the described deterioration of health in direct connection with the appointment at the sOC, which in some cases led to the termination of tests, the necessity to lie down, or even the lack of the possibility to leave the sOC independently. The participants in this study experienced the process of visiting the sOC as strenuous, overloading and exceeding the limits of their physical and mental resources. These can be classified as post-exertional malaise, which is considered a common symptom in LC [90] and one of the diagnostic criteria for ME/CFS [8,14,91,92]. Thus, there is a clear clinical and ethical implication to adapt care and treatment models to an individualized approach that takes into account functional limitations and resources [36,37,42,92,93]. In the sOC, the care model should be more tailored to the patient’s needs and resources to reduce or avoid overload. This study suggests the integration of telemedicine services seems to be an appropriate way to provide options to adapt the level of effort to the individual’s capacity. As described above, this can also be integrated into the collaboration between primary and specialty care. This aligns with Germany’s current long COVID guideline, a sub-statutory norm [94] and is identified by Chou et al. as a feature of LC care models [39]. Other options for taking into account the limitations and resources of patients in the sOC, based on the results of the study, could be to divide the appointment into several sessions on different days, to outsource certain examinations (e.g., ECG) to outpatient facilities, or to implement outreach care.

### 4.2. Strengths and Limitations

A strength of this cross-sectional exploratory study is that, owing to the theme-generating approach to data analysis, the diversity of the aspects identified and the sample size of 14 interviews, thematic data saturation can be assumed [61,62], and further interviews are not expected to identify any additional aspects in the target group. In addition, the willingness to participate was high, with almost 50% of those invited taking part in the study. In the German-speaking area, no further work on patient experiences in SOC is known; thus, explorative work can provide initial hypotheses in this context.

Methodological limitations arise from (1) the shift from purposive to convenience sampling, (2) the focus on a selected group of patients, namely, those who presented themselves at the sOC, and (3) the focus on one institution and one model of care. As a consequence of (1), it is possible that the results of the study are not representative of the care experiences of all sOC patients, but are primarily those of patients who had their appointment during the recruitment period, were interested in participating in the study, and met the inclusion criteria [60,95]. Based on (2), it was not possible to obtain the experiences of patients who felt well cared for in outpatient care or those who, owing to severe functional limitations, were unable to present themselves in specialized care. Due to (3), it is not possible at this time to say to what extent the concept of the sOC as described differs at other institutions in Germany. These limitations (1–3) mean that the respondents cannot be considered representative, and the results cannot be generalized, as common in exploratory qualitative design. Furthermore, selection bias may have occurred in that these patients may have agreed to participate in the study because they were satisfied with the appointment. The breadth of the results, with both negative and positive aspects, argues against this. Recall bias is possible due to the time lag between the sOC appointment and the interview appointment, as well as possible impairments caused by LC that affect memory [96]. The study only considered the patient’s perspective; therefore, it is not possible to assess how providers would describe the situation. The interview data were not triangulated with the questionnaire data in order to maintain the anonymity of the questions and to reduce the risk of drawing conclusions about an individual. This procedure means that the two sets of data stand alone, which could lead to a reduction in their informative value.

### 4.3. Implications for Research and Practice

#### 4.3.1. Implications for Research

Further health services research is needed to further evaluate and develop care provisions within and outside the sOC, which is supported by the recommendations of patient organizations [97]. The perspective of healthcare professionals involved should be added. The GPs’ and the sOC staff’s view of responsibility in the care process as well as the experience of the role can complement the patient’s perspective and thus allow a more comprehensive view, for example, considering possible effects on patient safety and quality of care. The potential impact of an appointment at the sOC on the further care process and patient-relevant outcomes should be further investigated in an extended study design, e.g., including quantitative surveys, and compared with regular outpatient care. Comparisons of different models of sOC services, including ambulatory models, and health economic evaluations should be added. The experiences with sOC care of LC patients with particularly severe symptoms should be explicitly added to tailor further development of care structures to this group as well.

#### 4.3.2. Implications for Practice

The results point to a clear need for action for care both inside and outside of the sOC; for example, patient narratives could be iteratively used to improve patient-centeredness, communication and quality of care and to empower patients and carers [98]. In addition, patients’ perspectives could be integrated into quality improvement methods such as quality improvement collaboratives [99], which have already been used in a UK study to improve LC care, offering opportunities to involve providers and patients in this change process [100]. The aforementioned approaches to cross-sector collaboration and consideration of appropriate resource allocation should be incorporated into the overall model.

## 5. Conclusions

The results of this study show that patients’ experiences at different points in the process of care in the sOC shape their expectations, experiences and anticipations of further care. This suggests that care in the sOC cannot be considered and designed in isolation but should be seen in the larger care context. Overall, the findings point to a need for adaptations in the outpatient long COVID clinic, such as identifying optimized models of care and tailoring them to the patient’s limited resources. This includes measures to support and improve general practitioner-based stepped care.

## Figures and Tables

**Table 1 healthcare-13-00818-t001:** Sociodemographic characteristics of the study participants—part 1 (n = 14).

	n (%)
Participants	14 (100)
Female	11 (78.6)
Male	3 (21.4)
Care during SARS-CoV-2-infection	
inpatient and outpatient	1 (7.1)
Outpatient	5 (35.7)
neither inpatient nor outpatient ^1^	8 (57.1)
Occupational situation	
Employed	12 (85.7)
unable to work	10 (71.4)
professional background in healthcare	4 (28.6)
Residential area	
rural residential area	5 (35.7)
urban residential area	9 (64.3)

^1^ meaning no contact with the healthcare system during COVID-19.

**Table 2 healthcare-13-00818-t002:** Sociodemographic characteristics of the study participants–part 2 (n = 14).

	Min	Max	Mean (SD)	Median
Age (years)	22	57	40.4 (11.8)	40.5
Time between COVID-19 and study participation (months)	4	29	10.4 (5.8)	10.0
Current functional scale ^1^	2	4	3.1 (0.6)	3.0
Functional scale before COVID-19	0	1	0.1 (0.3)	0.0

^1^ grade according to Klok et al. [63]; min. 0 ≙ no functional limitations, max. 4 ≙ severe functional limitations.

**Table 3 healthcare-13-00818-t003:** Themes and subthemes identified.

Themes	Subthemes
Pathway to the specialized long COVID outpatient clinic	Care outside the specialized long COVID outpatient clinic
	Own initiative and personal responsibility
	Appraisal of the stepped care approach
Care provided in the specialized long COVID outpatient clinic	Process of the appointment at the specialized long COVID outpatient clinic
	Experienced competences
	Differences to LC care outside of the specialized long COVID outpatient clinic
	Consequences of the appointment at the specialized long COVID outpatient clinic
	Single appointment at the specialized long COVID outpatient clinic
	Indications for an appointment at a specialized long COVID outpatient clinic

## Data Availability

The data presented in this study are available on request from the corresponding author. The datasets generated and/or analyzed during the current study are not publicly available because of assured data protection regulations. The raw data supporting the conclusions of this article will be made available by the corresponding author upon request.

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
