# Peer review of "Patient Perspectives on the Care in a Long COVID Outpatient Clinic—A Regional Qualitative Analysis from Germany"

_healthcare, 2025, doi:10.3390/healthcare13070818_

Round 1

Reviewer 1 Report

Comments and Suggestions for Authors

First of all, thank you for giving me the opportunity to evaluate the Manuscript. My views on the Manuscript are below.

Abstract Section,
- The abstract adequately reflects the content of the article and has been prepared in accordance with the rules.

Introduction Section,
- Although the introduction section is written in detail, a section on long coid-19 experiences was not sufficiently included. A paragraph stating patient experiences and the gap regarding this should be added.

Method Section,
- There is not enough information regarding the number of samples and when the sample will be terminated, a section regarding this should be added.
- What kind of interview form was used (it is understood from the table that it is structured, but it should be stated whether it is a structured or semi-structured form), whether expert opinion was obtained regarding the form and how the final questions were decided should be written.
- For qualitative research, the steps of internal and external validity and how these were achieved should be added (how were the records taken, with which tools were the records taken, how the interview was conducted, how many minutes did it take, etc.)
- While the analyses are written, it should be explained which analysis method was preferred and why. By what method were the themes determined (was there a conceptual framework, etc.).

Findings Section,
- The themes and sub-themes are explained in detail according to the given tables.

Discussion Section,
- Although the discussion is written sufficiently, it was not deemed appropriate to state convenience sample as a limitation in the limitations section. Purposeful sampling is used in these studies and there is no purpose such as generalization like quantitative research. Therefore, the limitations should be reviewed.

Author Response

Reviewer #1

EVALUATION

First of all, thank you for giving me the opportunity to evaluate the Manuscript. My views on the Manuscript are below.

Thank you very much for your valuable and constructive feedback. We hope that by integrating your suggestions, we have significantly improved the quality and informative value of the manuscript.

Below, please find our detailed revisions:

Abstract section

The abstract adequately reflects the content of the article and has been prepared in accordance with the rules.

Thank you very much for this feedback. Due to the concerns of another reviewer, we replaced the term “survey” with “qualitative analysis” (line 3) to the title to better describe the methodoligical approach.

Introduction section

Although the introduction section is written in detail, a section on long coid-19 experiences was not sufficiently included. A paragraph stating patient experiences and the gap regarding this should be added.

Thank you very much for your perspective.

We thoroughly considered your suggestion and decided to refrain from expanding on the patient experiences already mentioned in the introduction (lines 48-49 and 73-75) to reduce redundancies between the introduction and the discussion, as requested several times in the review.

Methods section

There is not enough information regarding the number of samples and when the sample will be terminated, a section regarding this should be added.

Thank you for your suggestion. We have now added more information on the termination of the sample and thoroughly revised the section on data saturation:

Recruitment was scheduled to stop when saturation could be assumed. Based on field notes, comparison of interview content aspects and literature on saturation and sample sizes in qualitative research [62,63]. Thematic saturation, thematic saturation was to be assumed when interviews had provided extensive information about the questions and aims of the study and only few new information could be collected through further interviews. Thus, and given the homogeneity of the study group, thematic saturation was assessed and recruitment was stopped on January 03, 2023.” (lines 113-126)

Detailed information about the sample, including the number of samples, is described in the results section under “3.1 Sample description”  (lines 202-215).

Methods section

What kind of interview form was used (it is understood from the table that it is structured, but it should be stated whether it is a structured or semi-structured form), whether expert opinion was obtained regarding the form and how the final questions were decided should be written.

Thank you for your valuable feedback.

To clarify we added the corresponding information in the description of the study design:

“A cross-sectional, exploratory qualitative study design [57,58] with problem-centered, semi-structured interviews [59] was chosen to collect the subjective and individual experiences of patients in as much detail and differentiation as possible.” (line 82-87)

We also added more information on how the final (wording of the) questions were decided:

“Based on the theoretical background and current research, a semi-structured interview guide was developed in an iterative  process of collecting, reviewing, sorting and subsuming appropriate questions [64,65]. Both, the thematic expertise of the first author, the methodological expertise of the second author, and the qualitative and clinical expertise of the last author were incorporated in this process. Owing to the high stress levels of the target group, a separate piloting of the interview guide was deemed challenging and potentially unethical. Therefore, the guide was piloted for comprehensibility and practicality in the first two interviews. Subsequently, in collaboration with the first and second author, the wording of few specific passages where comprehension issues arose in the interviews was slightly adjusted.“(lines 133-143)

Methods section

For qualitative research, the steps of internal and external validity and how these were achieved should be added (how were the records taken, with which tools were the records taken, how the interview was conducted, how many minutes did it take, etc.)

Thank you very much for this valuable feedback.

More details on the data collection were added:

“The participants individually chose the mode of interview based on their personal preference and capacities: telephone interview, video conference via the web conferencing service of Heidelberg University's computer center, face-to-face interview at the university hospital or in their personal home environment. Participants had the option of taking a break from the interview or continuing the interview on another day in order to be consistent with their individual capacity and limits and to avoid being overloaded by their participation in the study. No one other than the interviewer or the participant was present during the face-to-face interviews, and no one else was present during the telephone and video conference interviews, if this could be determined. In all the cases, field notes were taken, and the interviews were conducted by the first author recorded with a digital, portable voice recorder, transcribed verbatim, and pseudonymized f4 transkript Edu version 7.0.6 software in accordance with the institutional transcription guidelines. The transcripts were not submitted to the participants for correction or comment.” (lines 147-161)

Information on the duration and mode of the conducted interviews can be found under “3.1 Sample description” (lines 202-215).  In addition, we have revised the above sections to clarify the data saturation process. The process of reflexivity is described in lines 188-200.

Methods section

While the analyses are written, it should be explained which analysis method was preferred and why. By what method were the themes determined (was there a conceptual framework, etc.).

Thank you very much for this feedback.

The analysis method is described in the data analysis section. We added a justification for our approach and hope that this will answer your questions on the methods section:

“Reflexive thematic analysis [66] was chosen because of its flexibility allowing both inductive coding, in order to capture the patterned meaning of the largly unknown individual experiences of the patients with their care in the sOC, and deductive coding, to reflect the problem-centered nature of the interviews.” (lines 149-152)

Findings section

The themes and sub-themes are explained in detail according to the given tables.

Thank you very much for this positive feedback.

Discussion section

Although the discussion is written sufficiently, it was not deemed appropriate to state convenience sample as a limitation in the limitations section. Purposeful sampling is used in these studies and there is no purpose such as generalization like quantitative research. Therefore, the limitations should be reviewed.

Thank you for the suggestion. We have thoroughly revised the limitations section and also clarified the change from purposive to convenience, as well as the resulting restrictions.

Methodological limitations arise from 1) the shift from purposive to convenience sampling, 2) the focus on a selected group of patients, namely, those who presented themselves at the sOC, and 3) the focus on one institution and one model of care. As a consequence of 1), it is possible that the results of the study are not representative of the care ex-periences of all sOC patients, but are primarily those of patients who had their appoint-ment during the recruitment period, were interested in participating in the study, and met the inclusion criteria [61,95]. Based on 2), it was not possible to obtain the experiences of patients who felt well cared for in outpatient care or those who, owing to severe functional limitations, were unable to present themselves in specialized care. Due to 3), it is not pos-sible at this time to say to what extent the concept of the sOC as described differs at other institutions in Germany. These limitations (1-3) mean that the respondents cannot be con-sidered representative, and the results cannot be generalized, as common in exploratory qualitative design”. (lines 565-583)

Reviewer 2 Report

Comments and Suggestions for Authors

The manuscript provides valuable insights into patients' experiences in a specialized long COVID outpatient clinic (sOC) in Germany. The study addresses an important gap in understanding patients' perspectives on long COVID care, particularly in specialized clinical settings. The qualitative approach is appropriate to capture nuanced patient experiences. The findings make a meaningful contribution to the discussion on how to improve the health care system for patients with long COVID. However, some areas require further clarification, methodological refinement and discussion.

Major concerns:
1.    Methodological justification and sampling strategy:
The study initially used purposive sampling, but later switched to convenience sampling due to recruitment difficulties. While this is acknowledged, it introduces potential bias. The authors should discuss how this shift may have influenced the results and any limitations arising from it.
The sample size (n=14) appears appropriate for qualitative research, but could benefit from justification with references to saturation theory.
2.    Interpretation of findings:
The discussion section effectively links the findings to the literature, but could elaborate on potential solutions to improve the integration of primary and specialty care.
Given that patients reported worsening symptoms after their SOC appointment, the authors should discuss potential interventions to mitigate this problem (e.g., staggered appointments, alternative consultation methods such as telemedicine).
3.    Potential conflicts of interest:
Two authors were affiliated with the clinic studied. While conflicts of interest are declared, the manuscript would benefit from a statement of measures taken to minimize bias during data collection and analysis.

4.   Clarity of presentation:
 Some parts of the methodology section, particularly on data coding and thematic analysis, could be more explicitly described to ensure reproducibility.
The results section includes extensive patient quotes, which are valuable but could be presented more concisely and linked to the text.
5. Implications and future research:
The conclusion could be strengthened by emphasizing specific policy or practice recommendations, particularly regarding interdisciplinary collaboration and resource allocation.
Future research directions should include provider perspectives and comparative studies with other ambulatory models.

Conclusions: This study provides an insightful exploration of patient experiences in a long-term COVID outpatient clinic and offers valuable recommendations for improving models of care. Addressing the concerns outlined above will enhance the robustness and impact of the manuscript.

Author Response

Reviewer #2

EVALUATION

The manuscript provides valuable insights into patients' experiences in a specialized long COVID outpatient clinic (sOC) in Germany. The study addresses an important gap in understanding patients' perspectives on long COVID care, particularly in specialized clinical settings. The qualitative approach is appropriate to capture nuanced patient experiences. The findings make a meaningful contribution to the discussion on how to improve the health care system for patients with long COVID. However, some areas require further clarification, methodological refinement and discussion.

Thank you very much for your valuable and constructive feedback. We hope that by integrating your suggestions, we have significantly improved the quality and informative value of the manuscript.

Please find attached our detailed revisions:

1. Methodological justification and sampling strategy

The study initially used purposive sampling, but later switched to convenience sampling due to recruitment difficulties. While this is acknowledged, it introduces potential bias. The authors should discuss how this shift may have influenced the results and any limitations arising from it.
The sample size (n=14) appears appropriate for qualitative research, but could benefit from justification with references to saturation theory.

Thank you very much for your valuable and understanding feedback.

Regarding the shift from purposive to convenience sampling we adapted our discussion:

“Methodological limitations arise from 1) the shift from purposive to convenience sampling, 2) the focus on a selected group of patients, namely, those who presented themselves at the sOC, and 3) the focus on one institution and one model of care. As a consequence of 1), it is possible that the results of the study are not representative of the care experiences of all sOC patients, but are primarily those of patients who had their appointment during the recruitment period, were interested in participating in the study, and met the inclusion criteria [61,95]. Based on 2), it was not possible to obtain the experiences of patients who felt well cared for in outpatient care or those who, owing to severe functional limitations, were unable to present themselves in specialized care. Due to 3), it is not possible at this time to say to what extent the concept of the sOC as described differs at other institutions in Germany.  These limitations (1-3) mean that the respondents cannot be considered representative, and the results cannot be generalized, as common in exploratory qualitative design.” (lines 580-582)

We added the following to justify the assumed saturation:  “A strength of this cross-sectional exploratory study is that, owing to the theme-generating approach to data analysis, the diversity of the aspects identified and the sample size of 14 interviews, thematic data saturation can be assumed [62,94], and further interviews are not expected to identify any additional aspects in the target group.” (lines 558-561)

2. Interpretation of findings

The discussion section effectively links the findings to the literature, but could elaborate on potential solutions to improve the integration of primary and specialty care.
Given that patients reported worsening symptoms after their SOC appointment, the authors should discuss potential interventions to mitigate this problem (e.g., staggered appointments, alternative consultation methods such as telemedicine).

Thank you very much for your very important suggestions.

We added the following regarding the integration of primary and speciality care:

“Networks could constitute another approach to increase the competence and motivation of primary care physicians in LC care and promote cross-sectoral knowledge transfer and collaboration [18,89].  This could include, for example, telemedicine consultations or boards [39). Another supporting element is described by Stallmach et al. in terms of mobile primary healthcare for post-COVID patients in rural areas [90]. ” (lines 518-525)

We clarified the needs for adaption in line 533-537 “Thus, there is a clear clinical and ethical implication to adapt care and treatment models to an individualized approach”.

The telemedicine as a potential intervention has been extended and we added some other options:

“This study suggests the integration of telemedicine services seems to be an appropriate way to provide options to adapt the level of effort to the individual's capacity.   As described above, this can also be integrated into the collaboration between primary and specialty care. This aligns with Germany’s current long COVID guideline, a sub-statutory norm [94] and is identified by Chou et al. as a feature of LC care models [39]. Other options to take into ac-count the limitations and resources of patients in the sOC, based on the results of the study, could be to divide the appointment into several sessions on different days, to out-source certain examinations (e.g. ECG) to outpatient facilities, or to implement outreach care. (lines 542-555)

3. Potential conflicts of interest:

Two authors were affiliated with the clinic studied. While conflicts of interest are declared, the manuscript would benefit from a statement of measures taken to minimize bias during data collection and analysis.

Thank you very much for your criticism.

In order to minimize potential bias due to the conflicts of interest of those two authors, we made the following statement and have addressed another identified dual role.:

To minimize the development of biases, Sandra Stengel was not involved in data collection and not primary data analysis. During the study, she regularly reflected on her own assumptions, experiences and different roles in this context to be aware of various aspects influencing her perspective. During data collection and analysis, Uta Merle worked as a senior physician and Karin Tarbet worked as nurse in the institution whose patients' perspectives were being studied. Uta Merle and Karin Tarbet were not involved in data collection and in data analysis but only gave feedback and her consent on the conceptualization of the study and on the manuscript. Karin Tarbet also supported the recruitment of the study participants.” (lines 661-672)

4. Clarity of presentation:

Some parts of the methodology section, particularly on data coding and thematic analysis, could be more explicitly described to ensure reproducibility.
The results section includes extensive patient quotes, which are valuable but could be presented more concisely and linked to the text.

Thank you very much for this feedback.

Please refer to lines 157-169 which describe the data coding and thematic analysis. Together with the information below on the process of thematic analysis, we hope that you will be able to get a sufficient picture of our data analysis.

The six phases of thematic analysis (familiarizing with the data set; coding; generating initial themes; developing and reviewing themes; refining, defining, and naming themes; writing up) served as a guide for the data analysis, which was not seen as a linear process, but as an iterative one that could be adapted as needed.” (lines 171-174)

We have chosen not to shorten the quotes, as we want to report the statements in their entirety and consider this relevant to the understanding of the subject of the research.

5. Implications and future research

The conclusion could be strengthened by emphasizing specific policy or practice recommendations, particularly regarding interdisciplinary collaboration and resource allocation.
Future research directions should include provider perspectives and comparative studies with other ambulatory models.

Thank you very much for your suggestion.

As described under 2. Interpretation of findings, we added a paragraph on implications for practice:

“The aforementioned approaches to cross-sector collaboration and consideration of appropriate resource allocation should be incorporated into the overall model.” (lines 618-619)

We agree that future research should include the perspective of providers as we described it in the lines 660-661.

We added the important aspect of comparison with other ambulatory models:

“Comparisons of different models of sOC services, including ambulatory models, and health economic evaluations should be added.” (lines 607-608)

Reviewer 3 Report

Comments and Suggestions for Authors

Dear Authors,

Congratulations on the work you have developed! The chosen topic is highly relevant and timely, and the methodology used represents a valid approach for the qualitative research conducted. However, I believe some aspects could be improved to ensure greater clarity, depth, and impact:

  • The title could be more precise and objective. It is suggested to include a reference to "qualitative analysis" in the title to better reflect the qualitative approach used.
  • The abstract could be more structured, better highlighting the objectives, methodology, and main findings. Currently, it presents information in a somewhat generic manner.
  • Some sections, especially the discussion, contain lengthy passages and could be more concise to avoid redundancies.
  • In the methodology, the criteria for theoretical saturation are not clearly explained. Although the authors state that saturation was reached, there are not enough details on how this was determined.
  • The lack of data triangulation may be a limitation of the study and should be acknowledged in the limitations section.
  • The results section could be better structured to highlight the key findings more clearly. I suggest using more objective subheadings to help organize the reading.
  • The discussion repeats some information from the introduction, which could be avoided to reduce redundancies.
  • The clinical implications of the findings could be further explored. How can the results impact public policy formulation or the organization of Long COVID clinics?

By addressing these points, the study will achieve greater clarity and scientific robustness.

Author Response

Reviewer #3

EVALUATION

Dear Authors,

Congratulations on the work you have developed! The chosen topic is highly relevant and timely, and the methodology used represents a valid approach for the qualitative research conducted. However, I believe some aspects could be improved to ensure greater clarity, depth, and impact:

Thank you very much for your valuable and constructive feedback. We hope that by integrating your suggestions, we have significantly improved the quality and informative value of the manuscript.

Please find attached our detailed revisions:

·        The title could be more precise and objective. It is suggested to include a reference to "qualitative analysis" in the title to better reflect the qualitative approach used.

Thank you very much for this suggestion. We replaced the term “survey” with “qualitative analysis” to the title (line 3) to better describe the methodic approach.

·        The abstract could be more structured, better highlighting the objectives, methodology, and main findings. Currently, it presents information in a somewhat generic manner.

Thank you for your perspective. The structure of the abstract was adapted to the template of MDPI healthcare and the following headings were added: Background/Objectives, Methods, Results, and Conclusions (lines 11-22)

·        Some sections, especially the discussion, contain lengthy passages and could be more concise to avoid redundancies.

Thank you very much for your perspective.

We were able to significantly focus some sections of the discussion by making cuts.  Since the changes are spread over several sections, they are not listed in detail here.

·        In the methodology, the criteria for theoretical saturation are not clearly explained. Although the authors state that saturation was reached, there are not enough details on how this was determined.

Thank you very much for this constructive criticism.

As we don’t use Grounded Theory Methodology in this study, we don’t strictly apply the concept of theoretical saturation but referred more to data saturation. However, we agree that it is important to show how saturation was reached.

Therefore, we revised the section of data saturation as follows:

Recruitment was scheduled to stop when saturation could be assumed. Based on field notes, comparison of interview content aspects and literature on saturation and sample sizes in qualitative research [62,63]. Thematic saturation, thematic saturation was to be assumed when interviews had provided extensive information about the questions and aims of the study and only few new information could be collected through further interviews. Thus, and given the homogeneity of the study group, thematic saturation was assessed and recruitment was stopped on January 03, 2023.” (lines 146-152)

In addition, and in response to feedback from another reviewer, we have provided more details in the discussion section on the justification for the assumed saturation:

“A strength of this cross-sectional exploratory study is that, owing to the theme-generating approach to data analysis, the diversity of the aspects identified and the sample size of 14 interviews, thematic data saturation can be assumed [62,94], and further interviews are not expected to identify any additional aspects in the target group.” (lines 519-522)

·        The lack of data triangulation may be a limitation of the study and should be acknowledged in the limitations section.

Thank you for this feedback.

We added the following to the limitations section:

The interview data were not triangulated with the questionnaire data in order to maintain the anonymity of the questions and to reduce the risk of drawing conclusions about an individual. This procedure means that the two sets of data stand alone, which could lead to a reduction in their informative value.“ (lines 590-594)

·        The results section could be better structured to highlight the key findings more clearly. I suggest using more objective subheadings to help organize the reading.

Thank you again for your suggestion. We added the following subheadings to make reading easier:

·        Access to health care providers

·        Care experienced as inadequate

·        Openness to long COVID

·        Waiting time for the appointment at the specialized long COVID outpatient clinic

·        Course and duration of the appointment at the specialized long COVID outpatient clinic

·        Direct health consequences

·        Consequences for further care

·        Evaluation of one-time appointments at the specialized long COVID outpatient clinic

·        Possible reappointment at the specialized long COVID outpatient clinic

·        The discussion repeats some information from the introduction, which could be avoided to reduce redundancies.

Thank you very much for your perspective.

As reported above, we were able to significantly focus some sections of the discussion by making cuts.  Since the changes are spread over several sections, they are not listed in detail here.

·        The clinical implications of the findings could be further explored. How can the results impact public policy formulation or the organization of Long COVID clinics?

Thank you for this very important feedback. We added some suggestions on how the findings of the study might influence the development of public policy or the organization of Long COVID clinics:

 Networks could constitute another approach to increase the competence and motivation of primary care physicians in LC care and promote cross-sectoral knowledge transfer and collaboration [18,89]. This could include, for example, telemedicine consultations or boards [39]. A supporting element is described by Stallmach et al. in terms of mobile primary healthcare for post-COVID patients in rural areas [90].” (lines 518-523)

“In the sOC, the care model should be more tailored to the patient’s needs and resources to reduce or avoid overload. This study suggests the integration of telemedicine services seems to be an appropriate way to provide options to adapt the level of effort to the individual's capacity. As described above, this can also be integrated into the collaboration between primary and specialty care. This aligns with Germany’s current long COVID guideline, a sub-statutory norm [94] and is identified by Chou et al. as a feature of LC care models [39]. Other options to take into account the limitations and resources of patients in the sOC, based on the results of the study, could be to divide the appointment into several sessions on different days, to outsource certain examinations (e.g. ECG) to outpatient facilities, or to implement outreach care.” (lines 541-552)

“The aforementioned approaches to cross-sector collaboration and consideration of appropriate resource allocation should be incorporated into the overall model.” (lines 618-619)

Round 2

Reviewer 1 Report

Comments and Suggestions for Authors

The authors have made all requested edits.

Reviewer 2 Report

Comments and Suggestions for Authors

In my opinion, the article is good enough to be published.

Reviewer 3 Report

Comments and Suggestions for Authors

Congratulations on the work done! I believe the paper has improved in terms of rigor and clarity.